# Use of Polystyrene Nanoparticles as Collectors in the Flotation of Chalcopyrite

**DOI:** 10.3390/polym14235259

**Published:** 2022-12-02

**Authors:** Romina Murga, Camila Rodriguez, John Amalraj, Dennis Vega-Garcia, Leopoldo Gutierrez, Lina Uribe

**Affiliations:** 1Escuela de Ingeniería Civil de Minas, Universidad de Talca, Curicó 3340000, Chile; 2Centro de Recursos Hídrico para la Agricultura y la Minería (CRHIAM), Universidad de Concepción, Concepción 4030000, Chile; 3Instituto de Química de Recursos Naturales, Universidad de Talca, Talca 3480094, Chile; 4Departamento de Ingeniería Metalúrgica, Universidad de Concepción, Concepción 4030000, Chile

**Keywords:** nanoparticles, chalcopyrite microflotation, collector, polystyrene, copper sulfides

## Abstract

This study proposes the use of polymeric nanoparticles (NPs) as collectors for copper sulfide flotation. The experimental phase included the preparation of two types of polystyrene-based NPs: St-CTAB and St-CTAB-VI. These NPs were characterized by Fourier-Transform Infrared (FTIR) spectroscopy and dynamic light scattering (DLS). Then, microflotation tests with chalcopyrite under different pH conditions and nanoparticle dosages were carried out to verify their capabilities as chalcopyrite collectors. In addition, the zeta potential (ZP) measurements of chalcopyrite in the presence and absence of NPs were carried out to study their interaction. Lastly, some Atomic Force Micrographs (AFM) of NPs and Scanning Electronic Microscopy (SEM) and Energy Dispersive X-ray Spectroscopy (EDS) analysis of NPs on the chalcopyrite surface were conducted to analyze the size, the morphology and their interaction. The results obtained at pH 6 and pH 8 show that the NPs under study can achieve a chalcopyrite recovery near or higher than that obtained with the conventional collector. In this study, it was possible to observe that the NPs functionalized by the imidazole group (St-CTAB-VI) achieved better performance due to the presence of this group in its composition, allowing to achieve a greater affinity with the surface of the mineral.

## 1. Introduction

The production of copper from sulfide minerals represents more than 70% of the total copper production in Chile, contributing to increased ore processing by flotation plants [1]. It is important to note that, although froth flotation has been developed and commercially applied for more than a century, there are still some shortcomings that need to be overcome to achieve maximum mineral recovery, such as low grades of ores and environmental problems [2]. These shortcomings have become more challenging because the ore grades have consistently decreased, and the environmental awareness has become more relevant in recent years. More specifically, conventional reagents were classified as hazardous materials because they may affect both the environmental and human health as well as flora and fauna if they are improperly managed and disposed of [3,4]. Xanthates, for example, are the most common type of water-soluble collectors used for sulfide minerals, however, these have raised ever-increasing environmental concerns as carbon disulfide is readily emitted from xanthate, thus creating a need to search for a new and more efficient generation of flotation reagents [2,5,6,7].

Nowadays, one of the most interesting and promising alternatives to the use of conventional reagents is the use of nanoparticles (NPs). NPs appear as a potential alternative thanks to a series of extraordinary features, such as a large number of methods of synthesis, increased surface area, the ability to perform physical adsorption and easy detachment, recycling capabilities and low negative environmental impacts [2,8,9,10]. The potential of NPs as reagents has promoted substantial interest in flotation processes in recent years. There have been attempts to investigate hydrophobic nanoparticles as a new class of flotation collectors called “the core”, and to which functional groups are attached because of chemical reaction [6,9,11,12,13,14]. The functional nanostructures presented in the literature have cores that can be categorized in subgroups: polystyrene nanoparticles; cellulose nanocrystals/nanofibers and inorganic nanoparticles; and styrene and cellulose, which are distinguished by their ability to easily modify surface physicochemistry whilst inorganic nanoparticles are usually not chemically functionalized due to their structural nature [15].

Flotation studies demonstrated that hydrophobic nanoparticles may offer advantages as collectors in flotation over conventional collectors [6,8,16,17]. Researchers have suggested that as little as 10% coverage by nanoparticles on glass bead surfaces could promote high flotation efficiency, whereas a conventional molecular collector requires 25% or greater coverage for good recovery [6,8,14,17]. It has been demonstrated that water-soluble collectors can be replaced by hydrophobic polystyrene nanoparticles with the objective of increasing the separation efficiency of valuable minerals from the unwanted gangue materials, such the slime coating produced by phyllosilicates [11,12,17,18,19,20]. Different studies have proposed that nanoparticle collectors have the potential to give much more efficient attachment to air bubbles and to be less vulnerable to unwanted detachment than molecular collectors [12,13,18]. For example, it has been shown that most polystyrene nanoparticles will not detach from glass beads or mineral surfaces once deposited. This observation is important as nanoparticle collectors could function over a number of stages during a commercial flotation process, unlike conventional molecular collectors such as the xanthates [9].

The aim of this study was to evaluate the potential use of polymeric nanoparticles as flotation collectors at different pHs and correlate their properties with their ability to promote the flotation of copper sulfide minerals, considering two types of nanoparticles: polystyrene-based and imidazole-functionalized styrene nanoparticles. The second was chosen in order to evidence the action of ligand metal complexation of imidazole. The characterization by dynamic light scattering (DLS) and electrophoretic light scattering (ELS) were performed to establish their hydrodynamic diameter and determine the electrophoretic mobility, respectively. In addition, microflotation tests of chalcopyrite under different nanoparticle concentrations and pH values were carried out to compare recoveries with those obtained using potassium amyl xanthate. Lastly, the zeta potential (ZP) measurements of the NPs in the presence and absence of chalcopyrite were carried out to study their interaction and some Atomic Force Micrographs (AFM) of NPs and Scanning Electronic Microscopy (SEM) and Energy Dispersive X-ray Spectroscopy (EDS) analysis on the chalcopyrite surface were conducted to analyze the size and morphology and their interaction.

## 2. Materials and Methods

### 2.1. Materials

In the production of polymeric NPs, the following reagents were used: styrene (99%) as monomer, 1-vinylimidazole (VI, ≥99%) as ligand agent, cetyltrimethylammonium-bromide (CTAB, 99%) as cationic surfactants and emulsifiers and 2,2${}^{\prime }$-azobis (2-methylpropionamidine) dihydrochloride (V50, 99%) as promoter. These reagents were purchased from a commercial supplier (i.e., Sigma-Aldrich).

A high-purity chalcopyrite sample was supplied by Ward’s Natural Science and was used to evaluate the NPs as a collector of copper sulfides in the flotation process. The sample was ground using a planetary mono mill, Pulverisette 6 (Fritsch, Germany), to obtain a sample of chalcopyrite with a particle size in the range of −150/+75 μm for microflotation test and −75 μm for electrophoretic mobility measurements. The chemical composition of the sample was determined by X-ray fluorescence (XRF) with the Niton XL3 (Thermo-Fisher Scientific, Waltham, MA, USA), the mineralogical composition by X-ray Difractometry (XRD) using a MiniFlex 600 (Rigaku, Japan) and electrophoretic mobility measurements to determine the surface charge as a function of pH using Litesizer 500^®^ (Anton Paar, Austria) equipment.

Lastly, the flotation reagents considered for this study were the following: Aerofroth 70 supplied by SOLVAY, Antofagasta, Chile, as foaming agent. Hydrochloric acid (37%) and sodium hydroxide pellets (97%), and sodium chloride (99.99%) as modifier agents, purchased from MERCK and potassium amyl xanthate (PAX) supplied by ORICA (Santiago, Chile), were used as collector and previously purified according to the methodology described by Montalti [21]. In addition, it is important to note that all experiments were performed using 5 mM NaCl solutions which were prepared using Type II water (1 MW cm).

### 2.2. Polymerizations

Two types of polystyrene-based nanoparticles were prepared by emulsion polymerization. First, styrene and VI were purified by vacuum distillation while CTAB and V50 were used as supplied. The polymerization was conducted in a three-neck flask equipped with a condenser, two rubber stoppers holding syringe needles, and a magnetic stirring bar. One hundred grams of water was charged into the 250 mL three-neck round bottom flask and purged with nitrogen for 30 min prior to stirring (350 rpm). Then, 0.1 g CTAB (dissolved in 5 mL water) and 0.5 g of styrene were added. The mixtures were allowed to equilibrate for 10 min before 0.1 g of V50 was added to initiate the polymerization. After 15 min of the polymerization of the initial charge, an additional 4.5 g of styrene and 0.125 g of VI dissolved in 4.9 mL water were added over 5 h (0.0083 mL/min) from twin 10 mL syringes fitted to a syringe pump. The reaction was stirred at 70 °C for an additional 19 h. The resulting latex was dialyzed for at least one week against deionized water, after which the dialysate conductivity was less than 30 µS/cm. Once this process was finished, the NPs were denoted as St-CTAB-VI and St-CTAB. Table 1 shows the reagents and quantities used to prepare the different types of nanoparticles proposed in this study. It is important to note that the St-CTAB did not consider the addition of ligand agent VI.

Nanoparticle hydrodynamic diameters were determined by dynamic light scattering using an automatic detector angle, based on transmittance. The measurement time for each sample was set to 5 min. Electrophoretic mobility tests were performed at 25 °C in phase analysis light scattering mode. The reported EM values were of one run each 10 s. Dynamic light scattering and electrophoretic mobility measurements were performed using the Litesizer 500 and the Kalliope Software. Sample preparation included the dispersion of approximately 0.25 g/L of polystyrene nanoparticles in 5 mM NaCl in an omega cuvette.

### 2.3. Microflotation Test

A 140 mL Partridge–Smith glass cell was used to perform the micro-flotation tests. The assays were performed to evaluate the floatability of chalcopyrite in the presence of NPs and to compare them with the results obtained with a PAX collector. In these experiments, the chalcopyrite recovery was studied as a function of pH, the concentration of nanoparticles and the type of NPs. These experiments were conducted in duplicate for repeatability. PAX was studied at concentrations between 0 and 100 ppm, and St-CTAB-VI (23.65 mg/mL) and St-CTAB (24.91 mg/mL), considering additions between 0 and 4 mL, which corresponded to concentrations of 0–50 mg NPs/g chalcopyrite.

The procedure consisted in conditioning and flotation stages. The conditioning stage was initiated by adding a specific quantity of 5 mM NaCl solution in a beaker and adjusting the pH at the required value through the use of sodium hydroxide (NaOH) or hydrochloric acid (HCl), and stirring at 600 rpm. Then, 2 g of chalcopyrite and the required dose of NPs were added to solution, keeping the suspension on stirring for 5 min. Lastly, MIBC was added at a fixed concentration of 10 ppm, conditioning for 30 more seconds. Once this was achieved, the flotation process was carried out, and the pulp was transferred to the Partridge–Smith cell, where it was stirred at 900 rpm, and then the nitrogen was allowed into the cell (2 L/min). Flotation was performed for 2 min with manual scrapping every 10 s. Concentrates and tailings obtained in the tests were filtered and dried. The recovery was calculated considering the quotient between the concentrate mass and the feed mass. It is important to mention that the pH required was adjusted prior to the addition of NPs.

### 2.4. Zeta Potential

Zeta potential measurements were performed to study the interactions between chalcopyrite and NPs using the Litesizer 500^®^. A suspension was prepared by mixing 0.714 g of chalcopyrite (<75 μm) in 50 mL of 5 mM NaCl solution following the same procedure performed in the conditioning stage of microflotation tests and considering a nanoparticle and chalcopyrite concentration of approximately 0.25 g/L and 14.28 g/L, respectively. Once this process is completed, 0.4 mL of this suspension was transferred to an omega cuvette to determine the zeta potential. The measurement time was set to 5 min, with a fixed of temperature of 25 °C in phase analysis light scattering mode. Each condition was studied in triplicate to determine the standard deviation of these measurements.

### 2.5. Scanning Electron Microscopy with Energy Dispersive Spectroscopy (SEM/EDS)

The micrographs of nanoparticles were obtained by a LabRam HR800 Horiba AFM-RAMAN (Japan) and chalcopyrite and nanoparticles on chalcopyrite surface were obtained by a JEOL JSM-6380LV SEM (Japan) with an accelerating voltage of 25 kV, using the JEOL image processing software. Before starting the experiment, the edges and center of the slides were coated with silver coating and dried at 50 °C for 60 min. The samples were then coated under vacuum with a thin layer of 5 nm of platinum.

## 3. Results

### 3.1. Nanoparticle Characterization

Figure 1 shows the ST-CTAB-VI’s FTIR spectra, where in the peaks observed at around 3080, 3060 and 3025 cm${}^{-1}$ are attributed to the aromatic C-H stretches and the peaks found at 2925 and 2850 cm${}^{-1}$ are due to the asymmetric and symmetric stretches of CH${}_{2}$ groups, respectively. Two strong peaks observed at 756 and 700 cm${}^{-1}$ are attributed to the aromatic out-of-plane C-H bend and aromatic ring bend, respectively. This indicates that, indeed, the polymer was formed properly. Furthermore, the presence of the imidazole group in the nanoparticles is confirmed, which are generally found at approximately 1600, 1482 and 698 cm${}^{-1}$; these signals are attributed to the stretching of the bond C-N of the imidazole ring [9]. However, it is important to note that no major differences in FTIR spectra were observed because only 2.5 wt% of VI was used with respect to the styrene monomer and most of the strong bands of polystyrene and poly vinylimidazole are very similar. In addition, Figure 2 shows AFM images of St-CTA-VI (Figure 2a) and St-CTAB (Figure 2b); in these figures, it can be observed that that emulsion polymerization technique produced a uniform spherical shape formation of NPs with a particle size lower than 100 nm.

Table 2 presents a summary of the values obtained from the St-CTAB-VI and St-CTAB nanoparticles as a function of pH. These results show that St-CTAB had a larger average particle diameter and a higher polydispersity index than St-CTAB-Vi, indicating greater heterogeneity. This can be evidenced by observing that the values of D10, and D90 obtained in St-CTAB were far away from D50, in comparison with those values obtained in St-CTAB-VI. On the other hand, when comparing the particle sizes obtained at the different pHs, it was observed that the diameters of the different studied nanoparticles increased their value as a function of pH, obtaining the higher values at pH 8.

In addition, Table 2 shows the electrophoretic mobility of nanoparticles studied as a function of pH. The values obtained, show that both nanoparticles’ dispersions were positives for the whole pH range, which is an indication that the NPs are cations. This is due to the presence of amidine groups on the surface, which was added in all cases by the starter V50 and in some cases due to the presence of imidazole groups by the binding agent VI (St-CTAB-VI). However, it is important to note that electrophoretic mobilities were lower when the ionic strength was high, meaning a possible aggregation of nanoparticles when the pH increased. In addition, concerning the electrophoretic mobility magnitude, it is possible to note that the nanoparticles that presented lower electrophoretic mobility were St-CTAB-VI. This could be because these particles have fewer groups of surface charges per unit area due to their smaller hydrodynamic diameter.

### 3.2. Chalcopyrite Characterization

Table 3 shows the elemental composition of the ore obtained by X-ray fluorescence (XRF). The results indicate that the chalcopyrite sample was composed of 28.65%w copper, 30.08%w iron, 30.20%w sulfur and other elements, such as silicon, zinc and calcium below or near to 2.00%w. Furthermore, X-ray diffraction (XRD) analysis (Figure 3) showed that the ore was mainly composed of chalcopyrite (ICS/PDF-2-030289) and quartz (ICSD/PDF-2 036261). Lastly, a semi-quantification of crystalline phases was carried out using the Rietveld least squares method for the refinement of crystalline structures, which gave an adjustment parameter of Rwp = 10.65845%. This was considered adequate for the semi-quantification procedure. The percentages of each phase were corresponding to 97.76% of chalcopyrite and 2.24% of quartz.

In addition, Figure 4 shows the volume–weight particle size distribution of the chalcopyrite used for electrophoretic mobility and microflotation test. It can be observed that 55% of particles had a size of 35 μm in the cases of electrophoretic mobility, while in the microflotation test, 85% of particles had a size of 75 μm.

On the other hand, Table 4 presents the electrophoretic mobility and the zeta potential of chalcopyrite as a function of pH. The values show that copper mineral has a negative charge in the range of pH studied. This is consistent with the literature, considering that the point of zero charge (PZC) of this mineral is between pH 2 and pH 3 (Runqing et al. [22]).

### 3.3. Chalcopyrite Recovery at Different Dosages of NPs and Different pHs

Figure 5a–c shows the comparison of the recovery of chalcopyrite against the added doses of St-CTAB-VI, St-CTAB and PAX, at the different pHs studied. It can be seen from Figure 5a that in the lower range of concentration, with the increase in the dosage of St-CTAB-VI, the recovery increases gradually up to a peak, regardless of the pH value. After this peak value, the recovery decreases as the dosage of NPs keeps increasing. This is more evident for the experiment at pH 10. From the recovery curves obtained at natural pH, the best recovery of chalcopyrite was 98.07%, when 17.73 mg/g of NPs were used. On the other hand, comparing the curve at pH 8 with that obtained at natural pH, it is possible to observe that the maximum recovery was obtained at the same concentration of NPs, which corresponded to 93.67%. Finally, at pH 10, the recoveries were lower than those obtained at the other pHs studied, with a maximum recovery of 78.51% when 23.64 mg/g of NPs were used.

On the other hand, Figure 5b shows the comparison of the recovery of chalcopyrite against the added doses of St-CTAB at the different pHs studied. A similar trend to that obtained with St-CTAB-VI can be seen in Figure 6. That is, in a certain range, with the increase in the concentration of these NPs, the recovery increases gradually, regardless of the pH studied. However, at pH 8 and pH 10, it is observed that more dosages of NPs are required to reach a higher chalcopyrite recovery. From the recovery plots obtained, it can be observed that, at pH 6, the maximum recovery was 97.72%, when 18.62 mg/g of NPs were added. On the other hand, it is important to note that chalcopyrite recoveries obtained at pH 8 and pH were lower than those obtained at natural pH, reaching a maximum recovery of 87.37% (pH 8) and 76.90% (pH 10), when 24.91 and 31.14 mg/g of NPs were added. Lastly, at pH 10, it can be observed that, after obtaining the maximum chalcopyrite recovery, it started to decrease, when the concentration of nanoparticles increased.

Finally, Figure 5c shows the chalcopyrite recovery obtained with the collector PAX at the different values of pH studied. According to these results, it is possible to note that, at pH 6, the maximum recovery was near to 95% when 8.4 mg/g of PAX were used. Then, the recoveries obtained with NPs were better than those with the collector PAX. In addition, at pH 8, the recoveries with the different NPs studied were near to those obtained with PAX, considering that the recovery was 91.03% (8.4 mg/g PAX), achieving the best results with ST-CTAB-VI. Lastly, at pH 10, it can be observed that the collective action of the NPs was affected, obtaining a lower recovery in the presence of NPs than with PAX, considering that, in the presence of PAX, it was of 90.19%. Furthermore, it is important to note that even with higher concentrations of PAX, the chalcopyrite recovery was not significantly affected as in the case of higher concentrations of NPs. This could be related to the fact that the aggregation of NPs, which happens when high concentrations are used, could contribute to smaller chalcopyrite recoveries.

### 3.4. Zeta Potential Measurements

Table 5 shows results of the zeta potential measurements as a function of pH, considering the measures of the NPs alone (St-CTAB-VI and St-CTAB), chalcopyrite without NPs and in the presence of NPs (Chalcopyrite+St-CTAB-VI and Chalcopyrite+St-CTAB). According to the results presented in Table 5, it is possible to observe that the chalcopyrite has a negative potential which is between −5.79 ± 0.24 and −42.4 ± 0.93 mV, in the range of the pH studied. On the other hand, the NPs had a positive potential at the different pH studied, namely the St-CTAB with the greatest magnitude. After the interaction of mineral with the NPs, it can be observe that the zeta potential values of chalcopyrite change to positive values with magnitudes close to those of the nanoparticles, evidencing that, the interaction is due to the electrostatic forces considering that both NPs had cationic charges. In addition, the presence of the imidazole group in the St-CTAB-VI could be generating a more stable deposition of these NPs in the copper-rich surface, considering that the zeta potentials with the mixtures (Chalcopyrite+NPs) were near to those obtained in this type of NPs at the different pH studied.

### 3.5. SEM and XDS Analysis

Figure 6 presents SEM micrographs of chalcopyrite concentrate during the flotation test at pH 6 using St-CTAB-62. Figure 6 shows that, along with the collected chalcopyrite after flotation, very large nanoparticles aggregates were present with the chalcopyrite particles (spectra 19, 18, 20, 21). Although some of these may have formed during sample preparation, this image and the size distribution of NPs as a function of pH (Table 6) suggest significant nanoparticle aggregation. In addition, Table 6 shows that the EDS analysis performed for selected particles of chalcopyrite evidences that the particles that cover the chalcopyrite surface corresponded principally to the NPs due the high quantity of carbon/oxygen elements.

## 4. Discussion

The aim of this study was to evaluate the potential use of polystyrene-based nanoparticles as flotation collectors and correlate their properties with their ability to promote the flotation of copper sulfide minerals.

The NPs considered in this study were St-CTAB-VI and St-CTAB NPs, between which the main difference is the content of the co-monomer VI in the ST-CTAB-VI. The reagent VI is used to functionalize the NPs, as the presence of the imidazole as a functional group added more cationic charge additional to that provided by the CTAB itself. On the other hand, the V50 provides a cationic charge by the presence of the quaternary ammonium and amidine groups. This can be evidenced in the signals found at 1600, 1482 and 698 in St-CTAB-VI’s FTIR spectra, which are attributed to the stretching of the bond C-N of the imidazole ring.

The nanoparticle’s characterization showed that the emulsion polymerization produced uniform spherical NPs with a particle size lower than 100 nm, as the St-CTAB-VI is the smallest NP with the lowest polydispersity index, indicating that these were more monodisperse than St-CTAB. On the other hand, in relation to the electrophoretic mobility of the nanoparticles, it was possible to show that the different elaborated nanoparticles corresponded to cationic nanoparticles, which was expected due to the presence of amidine groups provided by the initiator V50 present in both types of NPs and the presence of imidazole groups provided by the VI ligand agent, only present in the St-CTAB-VI. In addition, regarding their magnitudes, it is possible to note from the electrophoretic mobility values in the range between 4.85 and 2.89 μm*cm/Vs that the St-CTAB-VI NPs are the ones with the lowest mobility. This behavior could be attributed to the smaller nanoparticles that carried nearly equivalent amounts of charged chemical groups that were distributed over a relatively larger surface area, resulting in fewer surface-charged groups per unit area. In addition, it is important to note that a lower variation on electrophoretic mobility as a function of pH was obtained for both types of NPs. This can be attributed to the presence of quaternary ammonium by the addition of CTAB, which displays a pH-independent degree of ionization.

From the microflotation tests, it was possible to note that, although the NPs concentration was between four and six times greater than the PAX concentration, the results obtained show that the NPs produced in this study can be used as a collector reagent for chalcopyrite because of the high recoveries obtained when 17.73 mg/g of St-CTAB-VI and St-CTAB were used at pH 6 and pH 8. Furthermore, it is important to note that St-CTAB-VI was more effective than the St-CTAB as a function of the different pH studied. This result is due to the fact that the presence of the imidazole group in the nanoparticle surface promotes their ability to selectively deposit onto the desired mineral particles; it is known that this group can form a ligand complex with copper ions. Considering the above, St-CTAB-VI could be even more selective for the copper minerals than the conventional collector (PAX). However, further studies related to the nano collector functionalization are needed in order to evidence its selectiveness, especially in the presence of clay minerals.

On the other hand, considering the chalcopyrite recovery obtained as a function of pH, it can be noted that when both the pH and the dosage of the reagent increased, the recovery of chalcopyrite was affected. To analyze the effect of pH on the hydrodynamic diameter of the NPs, it was observed that there was a considerable increase in the hydrodynamic size and the granulometric distribution of the nanoparticles at pH 8. This effect was more noticeable in the St-CTAB nanoparticles, which were characterized by presenting a higher PI index and a heterogeneous size distribution at the pHs studied. Furthermore, considering the zeta potential measures for NPs alone and chalcopyrite in the presence of NPs (Table 5), it can be noted that the chalcopyrite potential was completely overcompensated by the NPs and the St-CTAB nanoparticles have a higher surface charge than St-CTAB-VI, which may increase the probability of the free-surface charges to be oriented towards the aqueous media [23]. Therefore, this may produce chalcopyrite-nanoparticle agglomerates or lead to the hydration of the NP-coated mineral surfaces and then, making it hydrophilic when higher doses of NPs were used. Different authors argue that these increases in hydrodynamic diameter and particle size distribution as the pH increases are mainly associated with the occurrence of aggregation between nanoparticles, disfavoring the interaction with the mineral of interest [12,13,14,15,16,17,18,19]. In addition, these aggregates can contribute to reducing the kinetic rates of attachment between NP-deposited chalcopyrite and bubbles, and it is well known that more hydrophobic materials have faster kinetic rates with bubbles than less hydrophobic materials [16].

Considering the results of the experiments performed using different NPs, it could be noted that the chalcopyrite recovery obtained in the presence of St-CTAB is due to the deposition of cationic particles on the mineral surface by electrostatic and Van der Waals forces, while in the case of St-CTAB-VI, not only did these forces contribute to the aggregation of NPs, but also the presence of an imidazole group in the NPs provided to the formation of imidazole–copper complexation which generated a higher copper recovery. In addition, other important aspects to consider that conducted to the better response of ST-CTAB-VI than ST-CTAB as copper collector can be related to the fact that the NPs were smaller than St-CTAB. It is known that smaller nanoparticles will deposit themselves on the mineral of interest surface in less time, while larger particles could generate higher aggregation between them and generate a detachment on the mineral surface. Although the results were favorable in this study, further studies are required, including other types of copper minerals to evaluate its effectiveness as a collector for these minerals and considering the presence of pyrite (main gangue in the sulfurs flotation) to analyze its selectivity, and lastly, to evaluate whether its presence could help reduce the slime-coating produced by the presence of clay minerals, which affect the copper sulfide recovery.

## 5. Conclusions

From the results obtained in this study, it can be evidenced that cationic nanoparticles based on poly(styrene) with sizes smaller than 100 nm can interact with the copper sulfide mineral and induce its flotation between pH 6 and pH 10. In addition, to compare the recovery obtained with the collector PAX and the NPs at different pH values, it can be observed that the NPs obtained similar recoveries to those with the PAX reagent at pH 6 and pH 8. However, at pH 10, the collective effect of the NPs was affected, achieving lower recoveries than those obtained in the presence of PAX. Therefore, the NPs functionalized by the imidazole group (St-CTAB-VI) reached a better performance than simple cationic NPs (St-CTAB) due to the presence of VI in its composition that contributed to the formation of imidazole–copper complexation, which generated a higher chalcopyrite recovery. In addition, these NPs had the smallest hydrodynamic diameter, which allowed greater aggregation kinetics when interacting with the mineral of interest. 

## Figures and Tables

**Figure 1 polymers-14-05259-f001:**
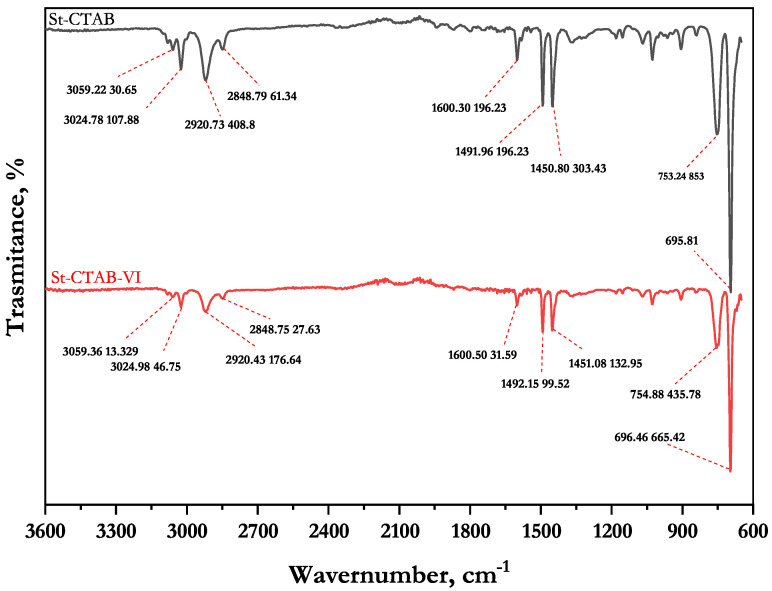
Fourier-Transform Infrared (FTIR) spectra of St-CTAB and St-CTAB-VI.

**Figure 2 polymers-14-05259-f002:**
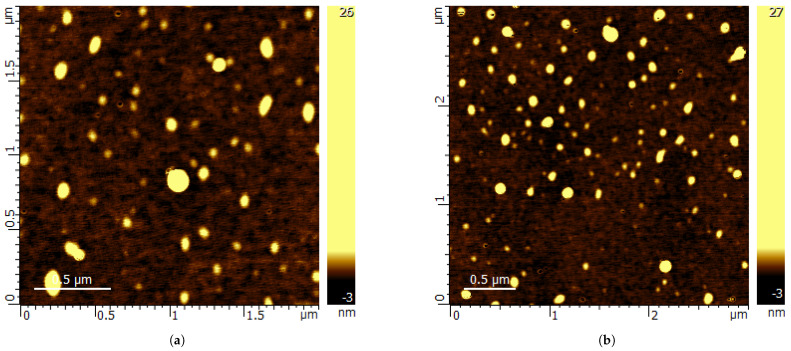
AFM micrographs of St-CTAB (**a**) and St-CTAB-VI (**b**).

**Figure 3 polymers-14-05259-f003:**
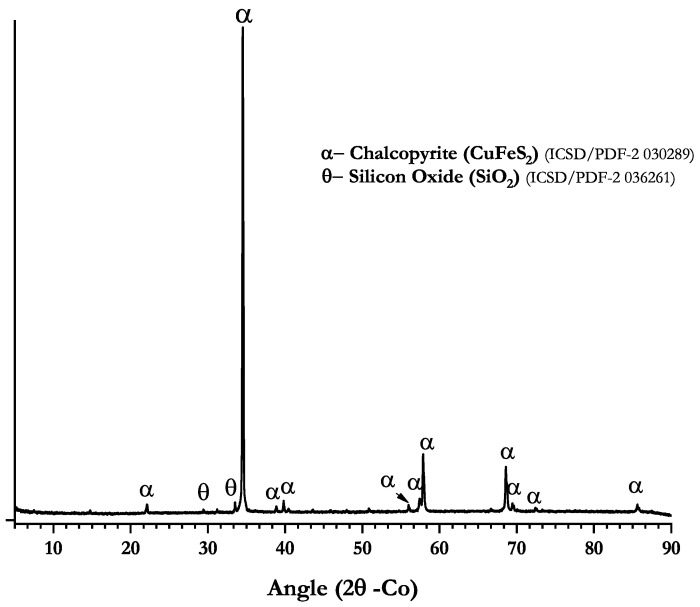
X-ray diffractogram of chalcopyrite sample.

**Figure 4 polymers-14-05259-f004:**
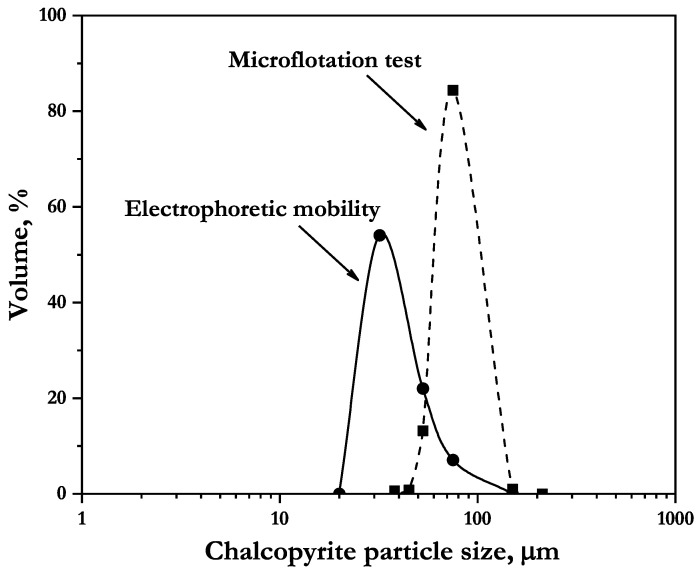
Particle size distribution of the chalcopyrite used for electrophoretic mobility and microflotation tests.

**Figure 5 polymers-14-05259-f005:**
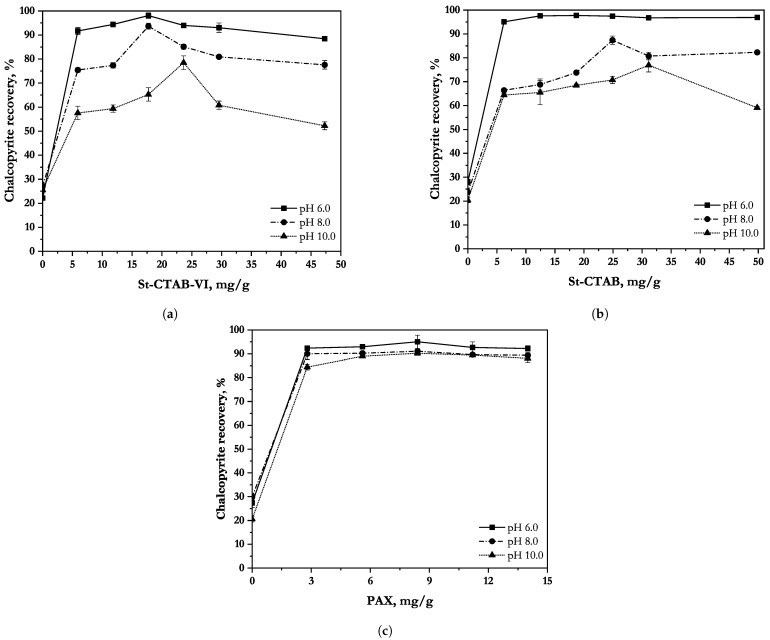
Chalcopyrite recovery as a function of St-CTAB-VI (**a**), St-CTAB (**b**) and PAX (**c**) concentration at different pHs, using 10 ppm of Aerofroth 70 and 5mM NaCl solution.

**Figure 6 polymers-14-05259-f006:**
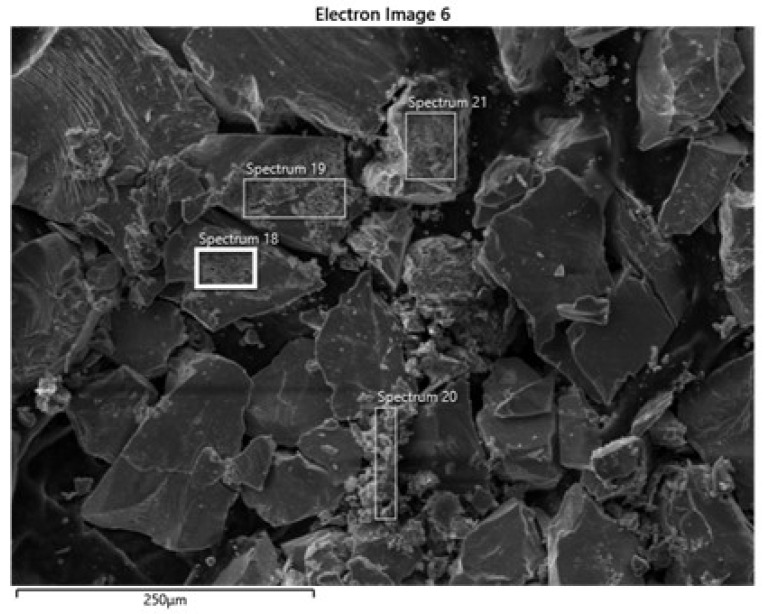
SEM micrographs of chalcopyrite collected as concentrated in the froth phase. The flotation employed 1.5 mL ST-CTAB and 2 g chalcopyrite, at pH 6 using 5 mM NaCl solution.

**Table 1 polymers-14-05259-t001:** Quantity of reagents used for the production of the NPs [9].

Name NPs	Initial Charge (g)				Polymerization (g)		
	Water	St	CTAB	V50	St	VI	Water
**St-CTAB-VI**	100	0.5	0.1	0.1	5.0	0.125	4.9
**St-CTAB**	100	0.5	0.1	0.1	5.0	-	-

**Table 2 polymers-14-05259-t002:** Hydrodynamic diameter, particle distribution, electrophoretic mobility and zeta potential of NPs.

Nanoparticle	pH	Hydrodynamic Diameter, nm (PI, %) *	$D_{10}$, nm	$D_{50}$, nm	$D_{90}$, nm	Electrophoretic Mobility, μm·cm/Vs	Zeta Potential, mV
**St-CTAB-VI**	6.0	49.69 (15.49)	30.34	47.16	73.40	3.21	42.61 ± 1.26
	8.0	78.36 (15.99)	47.41	72.17	109.25	3.05	39.16 ± 5.18
	10.0	74.29 (22.28)	36.26	53.18	87.14	2.89	37.02 ± 0.75
**St-CTAB**	6.0	61.74 (23.59)	32.35	40.71	130.00	4.84	62.08 ± 1.54
	8.0	85.55 (23.29)	44.21	78.23	156.28	4.70	60.30 ± 2.29
	10.0	62.61 (23.56)	33.34	59.66	132.00	4.45	57.10 ± 1.21

* PI: polydispersity index (%).

**Table 3 polymers-14-05259-t003:** X-ray fluorescence analysis for chalcopyrite ore.

Mineral Sample	Cu, %	Fe, %	Zn, %	Ca, %	Si, %	S, %
Chalcopyrite	28.65	30.08	1.03	1.14	1.96	30.20

**Table 4 polymers-14-05259-t004:** Electrophoretic mobility and zeta potential and for the chalcopyrite.

Mineral	pH	Electrophoretic Mobility, μm·cm/Vs	Zeta Potential, mV ± Std. Error
Chalcopyrite	6.0	−0.45	−5.79 ± 0.24
	8.0	−1.44	−18.43 ± 0.54
	10.0	−3.30	−42.47 ± 0.92

**Table 5 polymers-14-05259-t005:** Zeta potential measurements as a function of pH.

	Zeta Potential, mV		
	pH 6.0	pH 8.0	pH 10.0
Chalcopyrite	−5.79 ± 0.24	−18.44 ± 0.55	−42.39 ± 0.93
St-CTAB-VI	42.61 ± 1.26	39.16 ± 5.18	37.02 ± 0.75
St-CTAB	62.08 ± 1.54	60.30 ± 2.29	57.10 ± 1.21
Chalcopyrite + St-CTAB-VI	39.57 ± 0.60	38.94 ± 0.44	37.74 ± 0.46
Chalcopyrite + St-CTAB	41.30 ± 0.90	42.98 ± 0.99	40.97 ± 0.76

**Table 6 polymers-14-05259-t006:** EDS analysis done in selected particles of NPs.

Spectrums	Weight, %								
	S	Fe	Cu	C	O	Zn	Na	Al	Others
Spectrum 18	7.72	15.87	15.20	45.51	12.84	-	-	2.51	<1.00
Spectrum 19	12.98	10.28	10.34	49.49	15.58	-	-	-	<1.00
Spectrum 20	18.83	18.08	4.63	34.95	22.01	-	-	-	<1.00
Spectrum 21	15.70	2.25	2.10	43.12	10.50	22.72	2.46	-	<1.00

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
