# Peer review of "Use of Polystyrene Nanoparticles as Collectors in the Flotation of Chalcopyrite"

_polymers, 2022, doi:10.3390/polym14235259_

Round 1

Reviewer 1 Report

General Remarks

In this paper, two types of polymeric nanoparticles (NPS) are used to replace collectors in the flotation of chalcopyrite. Microflotation tests are performed using pure chalcopyrite particles, and surface characterization is done by using FTIR, SEM/EDS analysis, and zeta potential measurements. The aim of the study is to use the NPS as a flotation collector in copper flotation and compare their performance with potassium amyl xanthate (PAX), one of the most widely used collectors in sulfide ore flotation.   The question here is “Why?”. Flotation of copper minerals is a straightforward process and can be done effectively in many operations. Flotation of the secondary copper minerals and oxidized ore could be challenging to some extent. But specific collectors are developed by reagent manufacturers to solve this problem. Given that the purpose of using NPS, which requires specific reagents and production processes, should be stated clearly in this paper. Is it economic or environmental? It may not be increasing the efficiency of the flotation process, as it is mentioned above flotation of the copper minerals is a very well know and effective process on an industrial scale.

Besides, the adsorption mechanism of the NPS is based on electrostatic attraction on the mineral surfaces, and it is well known that the surface charge of the sulfide minerals and the non-sulfide gangue minerals (mostly quartz and feldspar) is similar and negative at a broad pH range (pH>2). In this case, how can the selectivity between these minerals be achieved?  

NPS type of reagents could be used as an alternative to the conventional collectors for ore types that can not be beneficiated with flotation. I think the copper sulfide ores are not in this group. Therefore, I think the contribution of this paper to the literature would be very limited.

Specific Remarks

The English of the paper is quite satisfactory in general, but there are some sections that need to be revised.

The title could be simpler like “Use of polystyrene nanoparticles as collectors in the flotation of chalcopyrite”

The microflotation results of PAX are not given. The optimum PAX dosage was given as 40 g/t on Page 9, and 60 ppm on Page 11.

The photo in Figure 2 is not clear.

Page 5: Line 166: There is something wrong with this sentence. It should be corrected.

Page 6: Line186: “.. had a size finer than..”

Page 7: Table 6: PZC of chalcopyrite is about pH 2-3 in the literature. In table 6, this value seems to be around pH=6.5. This happens when the surface of the sulfide minerals is oxidized, which means the chalcopyrite samples used in the tests do not represent freshly ground particles. More details should be given about the sample preparation and storage procedure followed in this study.

Page 8: Figure 5: The chalcopyrite recovery in the absence of a collector is very high, indicating a moderately oxidized surface. The collectorless flotation recovery was so high at pH 6, the use of NPS could barely increase the recovery. The results show that there is an optimum concentration window to get the maximum recovery with NPS, which is not the case for conventional collectors. Again, this comes to the question of the benefit of using NPS in chalcopyrite flotation.

Page 12: Line 306: The sentence should be revised.

Author Response

Dear reviewer 1, please refer to our answers below: 

  • Besides, the adsorption mechanism of the NPS is based on electrostatic attraction on the mineral surfaces, and it is well known that the surface charge of the sulfide minerals and the non-sulfide gangue minerals (mostly quartz and feldspar) is similar and negative at a broad pH range (pH>2). In this case, how can the selectivity between these minerals be achieved?

R/: This study is a first part of research proposes to find an alternative reagent to be used in copper sulfide minerals flotation with the advantages to be a collector less susceptible to the slime coating than traditional reagents in order to improve the recovery and the grade in the flotation process of low grade ores. The problem the industry is facing nowadays is that the grade of the ores is becoming more and more lower and this is affecting the performance of the operations (see slime coating for example). In addition, most of the reagents traditionally used for the flotation process have different environmental problems that needs to be taken into account. Please refer to the updated version of the manuscript for further information

  • NPS type of reagents could be used as an alternative to the conventional collectors for ore types that can not be beneficiated with flotation. I think the copper sulfide ores are not in this group. Therefore, I think the contribution of this paper to the literature would be very limited.

R/:The introduction section was improve in order to clarify ideas. Please refer to the updated version of the manuscript

  • The title could be simpler like “Use of polystyrene nanoparticles as collectors in the flotation of chalcopyrite”

R/: The title was modified. Please refer to the updated version of the manuscript.

  • The microflotation results of PAX are not given. The optimum PAX dosage was given as 40 g/t on Page 9, and 60 ppm on Page 11

R/: We had a typo in page 11. However, in the updated version of the manuscript was incorporated a new figure “chalcopyrite recovery as a function of PAX dosages at the different pH studied to clarify this comment

  • The photo in Figure 2 is not clear

R/: The photo was removed. Please refer to the updated version of the manuscript.

  • Page 5: Line 166: There is something wrong with this sentence. It should be corrected.

R/: The sentence was modified. Please refer to the updated version of the manuscript

  • Page 6: Line186: “.. had a size finer than..”

R/: The sentence was modified. Please refer to the updated version of the manuscript.

  • Page 7: Table 6: PZC of chalcopyrite is about pH 2-3 in the literature. In table 6, this value seems to be around pH=6.5. This happens when the surface of the sulfide minerals is oxidized, which means the chalcopyrite samples used in the tests do not represent freshly ground particles. More details should be given about the sample preparation and storage procedure followed in this study. (ROMINA)\

R/: We checked the PZ of chalcopyrite at pH 6 and the value was corrected. Please refer to the updated version of the manuscript.

  • Page 8: Figure 5: The chalcopyrite recovery in the absence of a collector is very high, indicating a moderately oxidized surface. The collectorless flotation recovery was so high at pH 6, the use of NPS could barely increase the recovery. The results show that there is an optimum concentration window to get the maximum recovery with NPS, which is not the case for conventional collectors. Again, this comes to the question of the benefit of using NPS in chalcopyrite flotation (ROMINA)

R/: You are correct, there was a big mistake to plot the chalcopyrite recovery in absence of collector at the different pH studied. Please refer to the updated version of the manuscript.

  • Page 12: Line 306: The sentence should be revised

The sentence was modified. Please refer to the updated version of the manuscript.

Thank you and best regards.

Reviewer 2 Report

This manuscript is set out to systematically investigate the polystyrene nanoparticle preparation and its utilization as the flotation collector of chalcopyrite. Some interesting results are obtained in this study but in my opinion, this article only provides superficial analysis for some complicated questions and the related explanations or discussions are not thorough enough. It is necessary for the authors to illustrate the research significance and advantages of novel polymeric collector. Please give a detailed elaboration about your investigation. Only some direct and simple conclusions are obtained from simple analysis methods. Here are some detailed comments:

1.     The corresponding analysis seems to be superficial and insufficiently verifiable, in which only Zeta potential and SEM micrographs were involved. Some more measurements like XPS and AFM should be provided to further study collector adsorption on the chalcopyrite surface.

2.     There is a lack of comparisons between polystyrene nanoparticles and conventional PAX collector except Table 7 of only three recovery values at different pH points.

3.     Why are there two kinds of polystyrene nanoparticles synthetized in this manuscript? What is its purpose and the flotation efficiency difference should be supplemented in more details. Please make some statements about this problem.

4.     Page 1, Line 16. There should be some literatures to support the sentence “The production of copper from sulphide minerals represents in Chile more than 70% of the total copper production.”

5.     Page 1, Line 19. The shortcomings of froth flotation should be expounded which needed to be overcome.

6.     Page 2, Line 65. The flotation particle in this study ranged from 75μm to 210μm. Is that too coarse and heavy to collect by flotation bubbles? Is there any references to support it? The granularity of -75μm is common to be adopted in flotation, there is a doubt here and please give an explanation.

7.     Page 2, Line 66. The abbreviation of X-ray fluorescence is not consistent here with that in Page 6, Line 176.

8.     Page 6, Line 180. The PDF cards of standard peaks in XRD analysis should be showed in Figure 3. Besides, the degree scale of left axis in Figure 3 should be deleted due to its arbitrary units.

9.     In Figures 5 and 6, when no collector was added, the natural flotation recovery of chalcopyrite at three pH values should be identical, but it is not here. Please give an explanation.

10.  The SEM micrograph resolution in this manuscript is a little low and not convenient for reading.

Author Response

Dear Reviewer 2, please refer to our answers below:

  1. The corresponding analysis seems to be superficial and insufficiently verifiable, in which only Zeta potential and SEM micrographs were involved. Some more measurements like XPS and AFM should be provided to further study collector adsorption on the chalcopyrite surface.

R/: Thank you for your observation, we will consider this comment in further studies.

  1. There is a lack of comparisons between polystyrene nanoparticles and conventional PAX collector except Table 7 of only three recovery values at different pH points. (ROMINA)

R/: A figure was added of chalcopyrite recovery as a function of PAX concentration. Please refer to the updated version of the manuscript.

  1. Why are there two kinds of polystyrene nanoparticles synthetized in this manuscript? What is its purpose and the flotation efficiency difference should be supplemented in more details. Please make some statements about this problem.

R/: two types of nanoparticles, polystyrene based and imidazole-functionalized styrene nanoparticles in order to evidence the action of surface ligand of imidazole to improve the selectivity in a next stage. Please refer to the updated version of the manuscript.

  1. Page 1, Line 16. There should be some literatures to support the sentence “The production of copper from sulphide minerals represents in Chile more than 70% of the total copper production.”

R/: The reference was added. Please refer to the updated version of the manuscript.

  1. Page 1, Line 19. The shortcomings of froth flotation should be expounded which needed to be overcome.

R/: It was already mentioned in the paper, however, some changes were made.

  1. Page 2, Line 65. The flotation particle in this study ranged from 75μm to 210μm. Is that too coarse and heavy to collect by flotation bubbles? Is there any references to support it? The granularity of -75μm is common to be adopted in flotation, there is a doubt here and please give an explanation.

R/: You are correct, there was a mistake the range was corrected. We usually use these range of size (see Powder technology. 2020, 359, 216-225). Please refer to the updated version of the manuscript.

  1. Page 2, Line 66. The abbreviation of X-ray fluorescence is not consistent here with that in Page 6, Line 176.

R/: It was corrected. Please refer to the updated version of the manuscript.

  1. Page 6, Line 180. The PDF cards of standard peaks in XRD analysis should be showed in Figure 3. Besides, the degree scale of left axis in Figure 3 should be deleted due to its arbitrary units.

R/: The PDF Card was added to the figure and the degree scale was delete. Please refer to the updated version of the manuscript.

  1. In Figures 5 and 6, when no collector was added, the natural flotation recovery of chalcopyrite at three pH values should be identical, but it is not here. Please give an explanation.

R/: You are correct, there was a big mistake to plot the chalcopyrite recovery in absence of collector at the different pH studied. Please refer to the updated version of the manuscript.

  1. The SEM micrograph resolution in this manuscript is a little low and not convenient for reading

R/: The photo was removed as the laboratory that helped us with the SEM micrographs did not send us a higher resolution image. Please refer to the updated version of the manuscript.

Thank you, best regards.

Reviewer 3 Report

Dear authors,

Thank you very much for the submission of your article to MDPI polymers. The study investigates the use of polymeric nanoparticles as collector of chalcopyrite. The synthesized nanoparticles were analyzed with respect to their chemical composition, granulometric properties and electric surface potential. Further, their efficiency as chalcopyrite collector was tested and the potential of the nanoparticles to render the electric surface potential of the mineral analyzed.

The topic fits well into the scope of the journal, but I have several questions  for which I would kindly ask the authors to respond on them, before a decision about the publication of the article can be done.

1.       The introduction requires more detailed information. The difference between water-soluble collectors (that are commonly used) and non-soluble nanoparticles as collectors has to be more explained and pronounced to make the special behavior of nanoparticles clear, i.e., the different adsorption mechanism and stability, the wetty-patch behavior (Ref. 11) or the random distribution of functional groups on the nanoparticle surface. In addition, the authors could also include recent findings on the use of renewable bio-polymers (cellulose, chitin, etc.) as collector and state the advantages and disadvantages of bio- and synthetic polymers (see ACS Sustainable Chem. Eng. 2022, 10, 32, 10570–10578, Langmuir 2021, 37, 2322−2333).

2.       The particle size distribution for chalcopyrite used in this study seems rather coarse. Especially for electrophoretic mobility tests, this size range is not applicable. Could the authors state more precisely the size fractions applied and maybe add a figure showing the distributions of the mineral used for the different techniques?

3.       In section 2.2, the second sentence is incomplete. Also the names of the samples is a bit weird. I can imagine that the authors have synthesized further samples that lead to the names of the samples, but for the sake of an easier reading of the article, the authors could denote simpler names for the two samples.

4.       State clear how many measurements on the electrophoretic mobility were dome for each sample, please.

5.       The value for the mass of the mass used in ZP measurements (line 127 0.714g) seems very arbitrary. Why would the authors use such an amount?

6.       Why are only the spectra for one sample shown in Figure 1? Please show also the spectra for the second sample and discuss them accordingly (presence of ligand VI or not).

7.       The quality of Figure 2 is not appropriate. Please add a figure that is more sharp.

8.       Table 2 as Figure 1, why only one sample? Please add the second, too.

9.       Please add the values for the zeta potential in Table 3. In addition, it would be interesting to analyze the concentration of amino groups on the nanoparticle surface.

10.   The values for EM shown in Table 6 are obtained from the coarse chalcopyrite sample? The stated size fraction is too coarse for EM measurements. Further, what equation was used to determine the zeta potential from the EM? Why would a higher value for EM (at pH 8) lead to a lower zeta potential (pH 10)?

11.   In Figure 5 and 6, the nit of the collector concentration should be stated as mg/g or similar. Further, even in the absence of collector the mineral shows high recoveries (>55%), which in consequence complicates the evaluation of the hydrophobization effect of the collector on the mineral surface. I suppose that N2 gas was used to avoid oxidation of the mineral surface and thus a decrease in the degree of hydrophobicity, but for practical considerations, the use of air or oxygen during flotation could have been more meaningful. Finally, how would the authors explain the drop of the recovery after reaching a maximum? This is especially interesting because at the highest collector concentration, recoveries equal or even lower than in the absence of collectors were achieved. Thus the question arises if the nanoparticles could act as depressants when too high concentrations are present (see Minerals Engineering 122 (2018) 44–52)?

12.   The comparison of the recovery using nanoparticles and PAX is not so straightforward. Why did the authors use a PAX concentration of 40 ppm and not higher? Therefore statements on “comparison” of water-soluble molecules and nanoparticles is always erroneous and have to be done with caution.

13.   The zeta potentials for chalcopyrite shown in Table 8 are very interesting, showing that either the electric surface potential was completely overcompensated by the nanoparticles, or that the chalcopyrite particle may have agglomerated and sedimentated, so that the device measured inly nanoparticles instead of chalcopyrite-nanoparticle agglomerates. Secondly, the mechanism of nanoparticle-chalcopyrite interaction is not answered properly. One could expect that negatively charged chalcopyrite interacts well with positively charged nanoparticles, but this is not the case under pH 6. Could the author state some ideas which surface sites of the nanoparticles could chemically interact with the mineral surface?

14.   The results shown in Table 8 could also be an answer for the decreasing flotation recoveries at higher collector concentrations, highlighting the different behavior of nanoparticles compared to amphiphilic molecules (see again Minerals Engineering 122 (2018) 44–52).

15.   In section Discussion, the authos state a PAX concentration of 60 ppm (line 289), instead 40 ppm stated previously in the Results section. Please check the used concentration.

16.   In the conclusion, the authors discuss solely an electrostatic interaction between the nanoparticles and chalcopyrite surface as main factor for the adsorption. However, at pH 6 both, the mineral and the nanoparticles are positively charged, that would lead to repulsive forces between them. Nevertheless, there is a flotation response at pH 6, which has to be explained by other adsorption mechanism.

Author Response

Dear reviewer 3,  please refer to our answers below:

  1. The introduction requires more detailed information. The difference between water-soluble collectors (that are commonly used) and non-soluble nanoparticles as collectors has to be more explained and pronounced to make the special behavior of nanoparticles clear, i.e., the different adsorption mechanism and stability, the wetty-patch behavior (Ref. 11) or the random distribution of functional groups on the nanoparticle surface. In addition, the authors could also include recent findings on the use of renewable bio-polymers (cellulose, chitin, etc.) as collector and state the advantages and disadvantages of bio- and synthetic polymers (see ACS Sustainable Chem. Eng.2022, 10, 32, 10570–10578, Langmuir 2021, 37, 2322−2333).

R/:The introduction section was improved, considering these comments. Please refer to the updated version of the manuscript.

  1. The particle size distribution for chalcopyrite used in this study seems rather coarse. Especially for electrophoretic mobility tests, this size range is not applicable. Could the authors state more precisely the size fractions applied and maybe add a figure showing the distributions of the mineral used for the different techniques?

R/: You  are correct. The size range of electrophoretic mobility test in the Litesizer 500 is between 38 nm and 100 µm. The size used for these measures was less to 75 µm. This information was added in the update version of the manuscript.

  1. In section 2.2, the second sentence is incomplete. Also the names of the samples is a bit weird. I can imagine that the authors have synthesized further samples that lead to the names of the samples, but for the sake of an easier reading of the article, the authors could denote simpler names for the two samples.

R/: The second sentence was completed and the names of samples were changed. Please refer to the updated version of the manuscript.

  1. State clear how many measurements on the electrophoretic mobility were dome for each sample, please

R/: We done three measurements on the electrophoretic mobility for each condition studied. We reported the average of the three measurements with the standard deviation. This information was added in the update version of manuscript.

  1. The value for the mass of the mass used in ZP measurements (line 127 0.714g) seems very arbitrary. Why would the authors use such an amount?

R/:the value for the mass used in ZP measurements was not arbitrary. This value was reduced because less quantity of samples is required for these measures. However, this value of mass stablished in ZP was calculated to maintain the same relation of chalcopyrite and NPS used in microflotation. This information was added in the update version of manuscript in order to clarify the process done.

  1. Why are only the spectra for one sample shown in Figure 1? Please show also the spectra for the second sample and discuss them accordingly (presence of ligand VI or not).

R/: The spectra for the second sample was incorporated and discussed.

  1. The quality of Figure 2 is not appropriate. Please add a figure that is more sharp.

R/: The photo was removed. Please refer to the updated version of the manuscript.

  1. Table 2 as Figure 1, why only one sample? Please add the second, too.

R/: The photo was removed. Please refer to the updated version of the manuscript.

  1. Please add the values for the zeta potential in Table 3. In addition, it would be interesting to analyze the concentration of amino groups on the nanoparticle surface.

R/: The zeta potential of NPS is in the Table 8, section zeta potential measurements to compare the ZP values obtained of NPS alone, Cpy without NPS and Cpy in the presence of NPS. According to you comment, in this section was added the analysis of the ZP measures obtained for NPS. Please refer to the updated version of the manuscript

  1. The values for EM shown in Table 6 are obtained from the coarse chalcopyrite sample? The stated size fraction is too coarse for EM measurements. Further, what equation was used to determine the zeta potential from the EM? Why would a higher value for EM (at pH 8) lead to a lower zeta potential (pH 10)?

R/: You are correct. The size used for these measures was less to 75 µm (The size range of electrophoretic mobility test in litesizer 500 is between 38 nm and 100 µm). The equation used to determine the zeta potential from the EM was the Smoluchowsky equation. Finally, in relation to the higher value for EM (at pH 8). There was a mistake write the values of EM for the pH 8 in the table. This information was added in the updated version of the manuscript.

  1. In Figure 5 and 6, the nit of the collector concentration should be stated as mg/g or similar. Further, even in the absence of collector the mineral shows high recoveries (>55%), which in consequence complicates the evaluation of the hydrophobization effect of the collector on the mineral surface. I suppose that N2gas was used to avoid oxidation of the mineral surface and thus a decrease in the degree of hydrophobicity, but for practical considerations, the use of air or oxygen during flotation could have been more meaningful. Finally, how would the authors explain the drop of the recovery after reaching a maximum? This is especially interesting because at the highest collector concentration, recoveries equal or even lower than in the absence of collectors were achieved. Thus the question arises if the nanoparticles could act as depressants when too high concentrations are present (see Minerals Engineering 122 (2018) 44–52)?

R/: You are correct, there was a big mistake to plot the chalcopyrite recovery in absence of collector at the different pH studied. Please refer to the updated version of the manuscript.

  1. The comparison of the recovery using nanoparticles and PAX is not so straightforward. Why did the authors use a PAX concentration of 40 ppm and not higher? Therefore statements on “comparison” of water-soluble molecules and nanoparticles is always erroneous and have to be done with caution.

R/: In the updated version of the manuscript was incorporated a new figure “chalcopyrite recovery as a function of PAX concentration at the different pH studied to clarify this comment.

  1. The zeta potentials for chalcopyrite shown in Table 8 are very interesting, showing that either the electric surface potential was completely overcompensated by the nanoparticles, or that the chalcopyrite particle may have agglomerated and sedimentated, so that the device measured inly nanoparticles instead of chalcopyrite-nanoparticle agglomerates. Secondly, the mechanism of nanoparticle-chalcopyrite interaction is not answered properly. One could expect that negatively charged chalcopyrite interacts well with positively charged nanoparticles, but this is not the case under pH 6. Could the author state some ideas which surface sites of the nanoparticles could chemically interact with the mineral surface?

R/: In the discussion section was consider this comment. Also, the PZ of chalcopyrite at pH 6 was wrong, because should be negative. We checked this result and was corrected. Please refer to the updated version of the manuscript.

  1. The results shown in Table 8 could also be an answer for the decreasing flotation recoveries at higher collector concentrations, highlighting the different behavior of nanoparticles compared to amphiphilic molecules (see again Minerals Engineering 122 (2018) 44–52).

R/:In the discussion section was consider this comment. Also it was added this reference. Please refer to the updated version of the manuscript

  1. In section Discussion, the authors state a PAX concentration of 60 ppm (line 289), instead 40 ppm stated previously in the Results section. Please check the used concentration.

R/: We had a mistake in line 289. However, in the updated version of the manuscript was incorporated a new figure “chalcopyrite recovery as a function of PAX dosages at the different pH studied in order to clarify this comment.

  1. In the conclusion, the authors discuss solely an electrostatic interaction between the nanoparticles and chalcopyrite surface as main factor for the adsorption. However, at pH 6 both, the mineral and the nanoparticles are positively charged, that would lead to repulsive forces between them. Nevertheless, there is a flotation response at pH 6, which has to be explained by other adsorption mechanism.

R/: The PZ of chalcopyrite at pH 6 was wrong, because should be negative. We checked this result and was corrected. Please refer to the updated version of the manuscript.

Thank you, best regards

Round 2

Reviewer 2 Report

Thank you for the authors. Even though many comments have been addressed, there are still some confusions needed to be explained in detail. I really keep confused about the significance of this study.

1.     Some more delicate measurements like XPS and AFM were still not be provided.

2.     The PDF cards have been supplemented, but their peaks were not shown in the figure.

3.     The description still seems not to be sufficient about the comparison between polystyrene nanoparticles and conventional PAX collector

Author Response

Dear Reviewer. Please refer to our answer below.

  1. Some more delicate measurements like XPS and AFM were still not be provided.

R/:  The AFM  images of NPS were provided. Please refer to the updated version of the manuscript.

  1. The PDF cards have been supplemented, but their peaks were not shown in the figure 

 R/:  The DRX image was improved. Please refer to the updated version of the manuscript.

  1. The description still seems not to be sufficient about the comparison between polystyrene nanoparticles and conventional PAX collector

R/: The paragraph associate to this comment was improved. Please refer to the updated version of the manuscript

Reviewer 3 Report

Dear authors,

Thank you very much for the submission of the revised article. Although the quality of the article has been improved during the first revision, I still have several questions that require consideration before the article can be accepted for publication

1.       The particle size distribution for chalcopyrite used for electrophoretic mobility tests is not stated! Please state more precisely the size fractions applied and maybe add a figure showing the distributions of the mineral used for the different techniques?

2.       The FTIR spectra for both samples is identical and there is no difference between the presence of VI or not. Why so?

3.       Please add the values for the zeta potential in Table 2 (in V instead of µm*cm/(Vs))). In addition, it would be interesting to analyze the concentration of amino groups on the nanoparticle surface.

4.       The explanation for the drop of the flotation recovery after reaching a maximum, when the nanoparticle collectors are used, is not properly explained in the article? Also the question if the nanoparticles could act as depressants, when too high concentrations are present, was not clarified properly (see Minerals Engineering 122 (2018) 44–52)?

5.       The mechanism of nanoparticle-chalcopyrite interaction is not answered properly. One could expect that negatively charged chalcopyrite interacts well with positively charged nanoparticles, but this is not the case under pH 6. Could the author state some ideas, which surface sites of the nanoparticles could chemically interact with the mineral surface, e.g. hydroxyl- or carboxyl- groups?

6.       In the conclusion, the authors discuss solely an electrostatic interaction between the nanoparticles and chalcopyrite surface as main factor for the adsorption. However, at pH 6 both, the mineral and the nanoparticles are positively charged, that would lead to repulsive forces between them. Nevertheless, there is a flotation response at pH 6, which has to be explained by other adsorption mechanism.

7.       Please edit the English spelling and grammar. There are still a lot of mistakes.

Author Response

Dear reviewer please refer to our answers below:

  1. The particle size distribution for chalcopyrite used for electrophoretic mobility tests is not stated! Please state more precisely the size fractions applied and maybe add a figure showing the distributions of the mineral used for the different techniques? 

R/: The particle size distribution for chalcopyrite for microflotation test and electrophoretic mobility were added. Please refer to the updated version of the manuscript.

  1. The FTIR spectra for both samples is identical and there is no difference between the presence of VI or not. Why so?

R/: We do not observe major differences in FTIR as we used only 2.5 wt% with respect to the styrene monomer in addition most of the strong bands of polystyrene and poly vinylimidazole are very similar. 

  1. Please add the values for the zeta potential in Table 2 (in V instead of µm*cm/(Vs))). In addition, it would be interesting to analyze the concentration of amino groups on the nanoparticle surface.

       R/:  The zeta potential values were added in Table 2

  1. The explanation for the drop of the flotation recovery after reaching a maximum, when the nanoparticle collectors are used, is not properly explained in the article? Also the question if the nanoparticles could act as depressants, when too high concentrations are present, was not clarified properly (see Minerals Engineering 122 (2018) 44–52)?

R/:  According this comment, discussion section was improved. Please refer to the updated version of the manuscript

  1. The mechanism of nanoparticle-chalcopyrite interaction is not answered properly. One could expect that negatively charged chalcopyrite interacts well with positively charged nanoparticles, but this is not the case under pH 6. Could the author state some ideas, which surface sites of the nanoparticles could chemically interact with the mineral surface, e.g. hydroxyl- or carboxyl- groups?

R/: In the first version of this manuscript, We had a mistake with the zeta potential of  chalcppyrite measure to pH 6. According to different references the PZ of Cpy at pH 6 should be negative. We checked the PZ of chalcopyrite at pH 6 and the value was corrected. ( -5,79+/- 0.24). However, we consider this comment and improve the analysis. Please refer to the updated version of the manuscript

  1. In the conclusion, the authors discuss solely an electrostatic interaction between the nanoparticles and chalcopyrite surface as main factor for the adsorption. However, at pH 6 both, the mineral and the nanoparticles are positively charged, that would lead to repulsive forces between them. Nevertheless, there is a flotation response at pH 6, which has to be explained by other adsorption mechanism.

R/: In the first version of this manuscript, We had a mistake with the zeta potential of chalcppyrite measure to pH 6. According. to different references the PZ of Cpy at pH 6 should be negative. We checked the PZ of chalcopyrite at pH 6 and the value was corrected. ( -5,79+/- 0.24). However, we consider this comment and improve the analysis. Please refer to the updated version of the manuscript

  1. Please edit the English spelling and grammar. There are still a lot of mistakes.

R/: The manuscript was checked, and some mistakes were corrected. Please refer to the updated version of the manuscript.

Round 3

Reviewer 2 Report

Thank you for the authors. Even though many comments have been addressed, there are still some confusions needed to be explained in detail. I really keep confused about the significance of this study.

1.     Some more delicate measurements like XPS and AFM were still not be provided.

2.     The PDF cards have been supplemented, but their peaks were not shown in the figure.

3.     The description still seems not to be sufficient about the comparison between polystyrene nanoparticles and conventional PAX collector